# Dimensionally Tight Bounds for Second-Order Hamiltonian Monte Carlo

**Oren Mangoubi**
EPFL
omangoubi@gmail.com

**Nisheeth K. Vishnoi**
EPFL
nisheeth.vishnoi@gmail.com

## Abstract

Hamiltonian Monte Carlo (HMC) is a widely deployed method to sample from high-dimensional distributions in Statistics and Machine learning. HMC is known to run very efficiently in practice and its popular second-order "leapfrog" implementation has long been conjectured to run in $d^{1/4}$ gradient evaluations. Here we show that this conjecture is true when sampling from strongly log-concave target distributions that satisfy a weak third-order regularity property associated with the input data. Our regularity condition is weaker than the Lipschitz Hessian property and allows us to show faster convergence bounds for a much larger class of distributions than would be possible with the usual Lipschitz Hessian constant alone. Important distributions that satisfy our regularity condition include posterior distributions used in Bayesian logistic regression for which the data satisfies an "incoherence" property. Our result compares favorably with the best available bounds for the class of strongly log-concave distributions, which grow like $d^{1/2}$ gradient evaluations with the dimension. Moreover, our simulations on synthetic data suggest that, when our regularity condition is satisfied, leapfrog HMC performs better than its competitors – both in terms of accuracy and in terms of the number of gradient evaluations it requires.

## 1 Introduction

Sampling problems are ubiquitous in a wide range of scientific and engineering disciplines and have received significant attention in Machine Learning and Statistics. In a typical sampling problem, one wants to generate samples from a given target distribution $\pi(x) \propto e^{-U(x)}$, where one is given access to a function $U : \mathbb{R}^d \to \mathbb{R}$ and possibly its gradient $\nabla U$. In many situations, such as when $d$ is large, sampling problems are computationally difficult, and Markov chain Monte Carlo (MCMC) algorithms are used. MCMC algorithms generate samples by running a Markov chain which converges to the target distribution $\pi$. Unfortunately, many MCMC algorithms work by taking independent steps of short size $\eta$, meaning that they typically only travel a distance roughly proportional to $\sqrt{i} \times \eta$ in $i$ steps, preventing the algorithm from quickly exploring the target distribution.

One MCMC algorithm that can take large steps is the Hamiltonian Monte Carlo (HMC) algorithm. Each step of the HMC Markov chain involves simulating the trajectory of a particle in the "potential well" $U$, with the trajectory determined by Hamilton's equations from classical mechanics [2, 38]. To ensure randomization, the momentum is refreshed after each step by independently sampling from a multivariate Gaussian. HMC is a natural approach to the sampling problem because Hamilton's equations preserve the target distribution $\pi$. This convenient property reduces the need for frequent Metropolis corrections which slow down traditional MCMC algorithms, and allows HMC to take large steps. HMC was first discovered by physicists [14], was adopted soon afterwards with much success in Bayesian Statistics and Machine learning [36, 37], and is currently the main algorithm used in the popular software package *Stan* [5]. Despite its popularity and the widespread belief that HMC is faster than its competitor algorithms in a wide range of high-dimensional sampling problems [11, 1, 38, 3], its theoretical properties are not as well-understood as its older competitor MCMC algorithms, such as the random walk Metropolis [35] or Langevin [16, 17, 13] algorithms. The lack of theoretical results makes it more difficult to tune the parameters of HMC, and prevents us

from having a good understanding of when HMC is faster than its competitor algorithms. Several recent papers have begun to bridge this gap, showing that HMC is geometrically ergodic for a large class of problems [30, 18] and proving quantitative bounds for the convergence rate of an idealized version of HMC on Gaussian target distributions [42]. Building on probablistic coupling techniques developed in [42], [16] and [13], [33] later proved a bound of $O^*(d^{\frac{1}{2}})$ gradient evaluations for a first-order implementation of HMC when $U$ is $m$-strongly convex with $M$-Lipschitz gradient (here the $O^*$ notation only includes dependence on $d$ and excludes regularity parameters such as $M, m$, accuracy parameters, and polylogarithmic factors of $d$). When the dimension is large, computing $O^*(d^{\frac{1}{2}})$ gradient evaluations can be prohibitively slow. For this reason, in practice it is much more common to use the second-order "leapfrog" implementation of HMC, which is conjectured to require $O^*(d^{\frac{1}{4}})$ gradient evaluations based on previous simulation [14] and asymptotic "optimal scaling" results [27, 39]. Very recently, [33] made some progress towards this conjecture by proving that $O^*(d^{\frac{1}{4}})$ gradient evaluations are required in the special case where $U$ is separable into orthogonal $O(1)$-dimensional strongly convex components satisfying Lipschitz gradient, Lipschitz Hessian and fourth-order regularity conditions.

**Our contributions.** We introduce a new, and much weaker, regularity condition that allows us to show that, in many cases, HMC requires at most $O^*(d^{\frac{1}{4}})$ gradient evaluations. Roughly, our regularity condition allows the Hessian to change quickly in "bad" directions associated with the data, while at the same time guaranteeing that the Hessian changes slowly in the directions traveled by the HMC chain with high probability (Assumption 1). The fact that our regularity condition need not hold for the "worst-case" directions allows us to show desired bounds on the number of gradient evaluations for a much larger class of distributions than would be possible with more conventional regularity conditions such as the Lipschitz Hessian property. Under our regularity condition we show bounds of $O^*(d^{\frac{1}{4}})$ gradient evaluations for the leapfrog implementation of HMC when sampling from a large class of strongly log-concave target distributions (Theorem 1). Next, we show that our regularity condition is satisfied by posterior distributions used in Bayesian logistic "ridge" regression. Computing these posterior distributions is important in statistics and Machine learning applications [40, 22, 32, 44, 25] and quantitative convergence bounds give insight into which MCMC algorithm to use for a given application, and how to optimally tune the algorithm's parameters. Finally, we perform simulations to evaluate the performance of the HMC algorithm analyzed in this paper, and show that its performance is competitive in both accuracy and speed with the Metropolis-adjusted version of HMC despite the lack of a Metropolis filter, when performing Bayesian logistic regression on synthetic data.

**Related work.** *Hamiltonian Monte Carlo.* The earliest theoretical analyses of HMC were the asymptotic "optimal scaling" results of [27], for the special case when the target distribution is a multivariate Gaussian. Specifically, they showed that the Metropolis-adjusted implementation of HMC with leapfrog integrator requires a numerical stepsize of $O^*(d^{-\frac{1}{4}})$ to maintain an $\Omega(1)$ Metropolis acceptance probability in the limit as the dimension $d \to \infty$. They then showed that for this choice of numerical stepsize the number of numerical steps HMC requires to obtain samples from Gaussian targets with a small autocorrelation is $O^*(d^{\frac{1}{4}})$ in the large-$d$ limit. More recently, [39] have extended their asymptotic analysis of the acceptance probability to more general classes of separable distributions.

The earliest non-asymptotic analysis of an HMC Markov chain was provided in [42] for an idealized version of HMC based on continuous Hamiltonian dynamics, in the special case of Gaussian target distributions. [33] show that idealized HMC can sample from general $m$-strongly logconcave target distributions with $M$-Lipschitz gradient in $\tilde{O}(\kappa^2)$ steps, where $\kappa := \frac{M}{m}$ (see also [4] for more recent work on idealized HMC). They also show that an unadjusted implementation of HMC with first-order discretization can sample with Wasserstein error $\varepsilon > 0$ in $\tilde{O}(d^{\frac{1}{2}}\kappa^{6.5}\tilde{\varepsilon}^{-1})$ gradient evaluations, where $\tilde{\varepsilon} := \varepsilon/\sqrt{M}$. In addition, they show that a second-order discretization of HMC can sample from separable target distributions in $\tilde{O}(d^{\frac{1}{4}}\tilde{\varepsilon}^{-1}f(m, M, B))$ gradient evaluations, where $f$ is an unknown (non-polynomial) function of $m, M, B$, if the operator norms of the first four Fréchet derivatives of the restriction of $U$ to the coordinate directions are bounded by $B$. [28] use the conductance method to show that an idealized version of the Riemannian variant of HMC (RHMC) has mixing time with total variation (TV) error $\varepsilon > 0$ of roughly $\tilde{O}(\frac{1}{\psi^2 T^2}R\log(\frac{1}{\varepsilon}))$, for any $0 \le T \le d^{-\frac{1}{4}}$, where $R$ is a regularity parameter for $U$ and $\psi$ is an isoperimetric constant for $\pi$.

*Langevin Algorithms.* [17] show that the unadjusted Langevin algorithm (ULA) can generate a sample from $\pi$ with TV error $\varepsilon > 0$ in $\tilde{O}(d\kappa^2\varepsilon^{-2})$ gradient evaluations. Using optimization-based techniques from [12], [7, 15] show bounds for ULA in KL divergence. [9] show that *underdamped* Langevin requires $\tilde{O}(d^{\frac{1}{2}}\kappa^2\tilde{\varepsilon}^{-1})$ gradient evaluations for Wasserstein error $\varepsilon > 0$ (see also [8]). [19] show that the Metropolis-adjusted Langevin algorithm (MALA) requires $\tilde{O}(\max(d\kappa, \ d^{\frac{1}{2}}\kappa^{1.5})\log(\frac{1}{\varepsilon}))$ gradient evaluations from a warm start in the TV metric.

*Hit and run, ball walk, Random walk Metropolis (RWM).* The Hit-and-run, ball walk, and RWM algorithms are all thought to have a step size of roughly $\Theta(1)$ on sufficiently regular target distributions [21]. Therefore, since most of the probability of a standard spherical Gaussian lies in a ball of radius $\sqrt{d}$, one would expect all three of these algorithms to take roughly $(\sqrt{d})^2 = d$ steps to explore a sufficiently regular target distribution. One should then be able to apply results such as [31] to show that, from a warm start, RWM requires $\tilde{O}(d\kappa\log(\frac{1}{\varepsilon}))$ target function evaluations to sample from the target distribution with TV error $\varepsilon$. Interestingly, the only result [19] we are aware of specialized for the strongly log-concave case gives a bound of $d^2\kappa^2\log(\frac{1}{\varepsilon})$ target function evaluations for RWM.

## 2  Hamilton's equations and the Hamiltonian Monte Carlo algorithm

In this section we present the background and HMC algorithm; see [38, 2] for a thorough treatment.

**Hamiltonian Dynamics.** A Hamiltonian of a simple system in $\mathbb{R}^d$ is $\mathcal{H}(q,p) = U(q) + \frac{1}{2}\|p\|_2^2$, where $q \in \mathbb{R}^d$ represents the "position" of a particle in this system, $p \in \mathbb{R}^d$ the "momentum," $U$ the "potential energy," and $\frac{1}{2}\|p\|_2^2$ the "kinetic energy." For fixed $\mathbf{q}, \mathbf{p} \in \mathbb{R}^d$, we denote by $\{q_t(\mathbf{q}, \mathbf{p})\}_{t\geq 0}$, $\{p_t(\mathbf{q}, \mathbf{p})\}_{t\geq 0}$ the solutions to Hamilton's equations:

$$\frac{dq_t(\mathbf{q}, \mathbf{p})}{dt} = p_t(\mathbf{q}, \mathbf{p}) \qquad \text{and} \qquad \frac{dp_t(\mathbf{q}, \mathbf{p})}{dt} = -\nabla U(q_t(\mathbf{q}, \mathbf{p})), \tag{1}$$

with initial conditions $q_0(\mathbf{q}, \mathbf{p}) = \mathbf{q}$ and $p_0(\mathbf{q}, \mathbf{p}) = \mathbf{p}$. When the initial conditions $(\mathbf{q}, \mathbf{p})$ are clear from the context, we write $q_t$, $p_t$ in place of $q_t(\mathbf{q}, \mathbf{p})$ and $p_t(\mathbf{q}, \mathbf{p})$. The gradient $-\nabla U$ in the second Hamilton equation is thought of as a "force" which acts on the particle.

**HMC.** We first consider an idealized version of the HMC Markov chain $X_0, X_1, \ldots$ based on the continuous Hamiltonian dynamics, with update rule $X_{i+1} = q_T(X_i, \mathbf{p}_i)$, where $\mathbf{p}_1, \mathbf{p}_2, \ldots \sim N(0, I_d)$ are iid. Since solutions to Hamilton's equations have invariant distribution $\propto e^{-\mathcal{H}(q,p)} = e^{-U(q)}e^{-\frac{1}{2}\|p\|_2^2}$, idealized HMC has stationary distribution $\pi(q) \propto e^{-U(q)}$ equal to the target distribution, without needing a correction such as Metropolis adjustment. This allows HMC to take much larger steps, and hence mix faster, than would otherwise be possible. It is not possible to implement an HMC Markov chain with continuous trajectories, so one must discretize these trajectories using a numerical integrator, such as the popular second-order leapfrog integrator (Step 5 in the algorithm below). In this case, one obtains the following *unadjusted* HMC (UHMC) Markov chain. The number of gradient evaluations required by UHMC is the main object of study in this paper.

---

**Algorithm 1** Unadjusted HMC

**input:** Initial point $X_0^\dagger \in \mathbb{R}^d$, oracle for gradient $\nabla U$, $T > 0$, $i_{\max} \in \mathbb{N}$, discretization level $\eta > 0$
**output:** Samples $X_0^\dagger, \ldots, X_{i_{\max}}^\dagger$ from the (following) UHMC Markov chain

1: **for** $i = 0$ to $i_{\max} - 1$ **do**
2:    Sample $\mathbf{p}_i \sim N(0, I_d)$
3:    Set $\mathrm{q}_0 = X_i^\dagger$ and $\mathrm{p}_0 = \mathbf{p}_i$
4:      **for** $j = 0$ to $\lfloor\frac{T}{\eta}\rfloor - 1$ **do**
5:    Set $\mathrm{q}_{j+1} = \mathrm{q}_j + \eta\mathrm{p}_j - \frac{1}{2}\eta^2\nabla U(\mathrm{q}_j), \qquad \mathrm{p}_{j+1} = \mathrm{p}_j - \frac{1}{2}\eta\nabla U(\mathrm{q}_j) - \frac{1}{2}\eta\nabla U(\mathrm{q}_{j+1})$
6:      **end for**
7:    Set $X_{i+1}^\dagger = \mathrm{q}_{\lfloor\frac{T}{\eta}\rfloor}$
8: **end for**

---

**Initialization:** In this paper we prove gradient evaluation bounds for UHMC from both a *warm start* and *cold start*, which we define as follows:

**Definition 1.** *(Warm start) Let $X_0, X_1, \ldots$ be a Markov chain, and let $\pi$ be our target distribution. We say that $X$ has an $(\omega, \hat{\delta})$-warm start if there is a random variable $\tilde{Y}_0 \sim \pi$ such that $\|X_0 - \tilde{Y}_0\|_2 < \omega/\sqrt{M}$ with probability $1 - \hat{\delta}$ for some $\omega, \hat{\delta} > 0$.*

**Definition 2.** *(Cold start) We say that $X$ has a cold start if $X_0 = x^\star$, with $x^\star := \operatorname{argmin}_{x \in \mathbb{R}^d} U(x)$.*

Since UHMC requires $\Theta(\frac{T}{\eta})$ gradient evaluations to compute each Markov chain step $i$, the total number of gradient evaluations required by UHMC is $\Theta(i_{\max} \times \frac{T}{\eta})$. Note that the parameters $i_{\max}$, $T$, $\eta$ are chosen by the user, and the optimal choice of these algorithm parameters may depend on the dimension $d$ and the regularity parameters of $U$ such as $M$ and $m$.

**Remark 1.** *The number of arithmetic operations required to compute the gradient depends on how the function $U$ is provided to us in a given application. In the Bayesian logistic regression application analyzed at the end of Section 4 of this paper, the number of arithmetic operations required to compute the gradient $\nabla U$ is $\Theta(d)$, and is the same number of operations required to evaluate the target function $U$ itself. However, in other applications it can take $2d$ times as many arithmetic operations to compute the gradient $\nabla U$ as it takes to compute the target function $U$.*

## 3 Regularity conditions

In this section we explain the $\sqrt{d}$ gradient evaluation bound barrier in prior approaches and present our regularity condition that overcomes it. Let $H_x$ denote the Hessian of $U$ at $x \in \mathbb{R}^d$. We start by noting that if one attempts to bound the number of gradient evaluations required by HMC using a conventional Lipschitz bound on the Hessian

$$\|(H_y - H_x)p\|_2 \le L_2 \|y - x\|_2 \times \|p\|_2 \qquad \forall x, y, p \in \mathbb{R}^d \tag{2}$$

that is defined with respect to the Euclidean norm, then the bounds that one obtains are no faster than $\sqrt{d}$ gradient evaluations. The reason is that if we use the usual "Euclidean" Lipschitz Hessian condition to bound the numerical error, we obtain an error bound of roughly $\sqrt{d}$, since (from a warm start) the trajectories of HMC travel with momentum roughly $N(0, I_d)$, implying that the momentum of these trajectories has Euclidean norm $\sqrt{d}$ with high probability (w.h.p.). To bound the error of a second-order method such as the leapfrog method used by HMC, we must bound the change of the directional derivative of the gradient along the path taken by the trajectories of the Markov chain. In particular, when the leapfrog integrator (step 5 of Algorithm 1) takes a numerical step from $q_j$ to roughly $q_{j+1} \approx q_j + \eta p_j$, one component of the error in computing the continuous Hamiltonian trajectory can be bounded by the quantity $\|(\eta^2 H_{q_j + \eta p_j} - \eta^2 H_{q_j})p_j\|_2$. This quantity in turn can be bounded using the Lipschitz Hessian constant by $\eta^2 L_2 \|\eta p_j\|_2 \times \|p_j\|_2$. Since $p_j$ is roughly $N(0, I_d)$ we have $\|p_j\|_2 \approx \sqrt{d}$ w.h.p., which gives an error bound of $\eta^3 L_2 d$ for one leapfrog step and roughly $\eta^2 L_2 d$ for the error of computing an entire HMC trajectory if $T = \Theta^*(1)$. To obtain an error bound of $\varepsilon$ we therefore need $\eta = O^*(1/\sqrt{d})$. When computing a trajectory of length $\Theta^*(1)$ with this stepsize $\eta$, we therefore need to compute $O^*(\sqrt{d})$ numerical steps. To overcome this $\sqrt{d}$ gradient evaluation barrier, we therefore need to control the change in the Hessian with respect to a norm which does not grow as quickly with the dimension as the Euclidean norm for a random $N(0, I_d)$ momentum vector.

We need a better way to bound the quantity $\|(\eta^2 H_{q_j + \eta p_j} - \eta^2 H_{q_j})p_j\|_2$. One way to do so would be to replace the Euclidean Lipschitz Hessian condition with an infinity-norm Lipschitz condition $\|(H_y - H_x)v\|_2 \le L_\infty \times \|y - x\|_\infty \|v\|_\infty$ for some constant $L_\infty > 0$. For this norm, $\|p_j\|_\infty = O(\log(d))$ with high probability since, roughly, $p_j \sim N(0, I_d)$, implying that $\|(\eta^2 H_{q_j + \eta p_j} - \eta^2 H_{q_j})p_j\|_2$ is bounded by roughly $\eta^2 L_\infty \log(d)$ rather than $\eta^2 L_2 d$.

Since for many distributions of interest this condition does not hold for a small value of $L_\infty$, we generalize this condition, to obtain a smaller $L_\infty$ constant for a wider class of distributions. Towards this end, we define the vector (semi-)norm $\|\cdot\|_{\infty,u}$ with respect to the collection of unit vectors $u := \{u_1, \ldots, u_r\}$ by $\|x\|_{\infty,u} := \max_{i \in \{1, \ldots, r\}} |u_i^\top x|$. The usual infinity norm is just a special case of this new norm if we set $u_i = e_i$ to be the coordinate vectors. Under this more general norm, the magnitude of a random $N(0, I_d)$ vector still grows only logarithmically with $d$, since each component $u_i^\top x$ is a univariate standard normal. The associated matrix norm $\|A\|_{\infty,u}$ is defined to be $\sup_{\|x\|_{\infty,u} \le 1} \|Ax\|_2$. Using this norm, and motivated by the discussion above, we arrive at our new regularity condition. Roughly speaking, our new regularity condition allows the Hessian to change very quickly in $r > 0$ "bad" directions $u_1, \ldots, u_r$, as long as it does not change quickly on average in a random direction.

**Assumption 1 (Infinity-norm Lipschitz condition).** *There exist $L_\infty > 0$, $r \in \mathbb{N}$, and a collection of unit vectors $u = \{u_1, \ldots, u_r\} \subseteq \mathbb{S}^d$, such that $\|H_y - H_x\|_{\infty,u} \le L_\infty \sqrt{r}\|y - x\|_{\infty,u}$ for all $x, y \in \mathbb{R}^d$.*

We expect this assumption to hold when the target function $U$ is of the form $U(x) = \sum_{i=1}^{r} f_i(u_i^\top x)$ for functions $f_i : \mathbb{R} \to \mathbb{R}$ with uniformly bounded third derivatives. In particular, this class includes the target functions used in logistic regression. This condition may also be of independent interest.

**Additional conditions for cold starts.** When proving bounds from a cold start, roughly speaking we would still like to guarantee that the HMC trajectories travel with speed $O^*(1)$ in any of the "bad" directions, so that $\|p_t\|_{\infty,\mathsf{u}} = O^*(1)$. However, unlike from a warm start, we have no guarantee that the momentum is roughly $N(0, I_d)$. To bound $\|p_t\|_{\infty,\mathsf{u}}$ we therefore need another way to control the growth of the quantity $|u_i^\top p_t|$ in each "bad" direction $u_i$. To do so, we would like to guarantee that the bounds on the "force" acting on our Hamiltonian trajectory in each $u_i$ direction depend only on the components $u_i^\top q_t$ and $u_i^\top p_t$ of the position and momentum in that direction, regardless of the component of the momentum orthogonal to $u_i$. Towards this end, we assume the following:

**Assumption 2 (Gaussian tail bound condition (for cold start only)).** *There exists a constant $b > 0$, and a collection of unit vectors $\mathsf{u} = \{u_1, \ldots, u_r\} \subseteq \mathbb{S}^d$, such that $\min\{mu^\top(x-x^\star), Mu^\top(x-x^\star)\} - b \leq u^\top \nabla U(x) \leq \max\{mu^\top(x-x^\star), Mu^\top(x-x^\star)\} + b$ for all $x \in \mathbb{R}^d$, $u \in \mathsf{u}$.*

Assumption 2 gaurantees that the component of the gradient in each "bad" direction $u_i$ is bounded solely in terms of the component of the position in that same "bad" direction. This allows us to apply arguments based on Gronwall's inequality on the projection of the trajectory in each bad direction $u_i$ in order to bound the magnitude of the position and momentum at time $t$ in the direction $u_i$. Using Grownwall's inequality, we bound the component of the initial position and momentum in the direction $u_i$, without assuming a warm start.

# 4 Theoretical results

Our main result is a bound on the number of gradient evaluations required by HMC with second-order leapfrog integrator under the infinity-norm Lipschitz condition (Assumption 1), when sampling from $\pi(x) \propto e^{-U(x)}$ if $U$ is $m$-strongly convex with $M$-Lipschitz gradient. We bound the required number of gradient evaluations for both a warm and cold start. Here we focus on the warm start result; see the arXiv version [34] for bounds from a cold start and a more formal statement of our warm start result.

**Theorem 1 (Bounds for second-order HMC, informal).** *Let $\pi(x) \propto e^{-U(x)}$ where $U : \mathbb{R}^d \to \mathbb{R}$ is $m$-strongly convex, $M$-gradient Lipschitz, and satisfies Assumption 1. Then there exist parameters $T, \eta, i_{\max}$, such that from an $(\omega, \delta)$-warm start, Algorithm 1 generates an approximate independent sample $X_{i_{\max}}^\dagger$ from $\pi$ such that $\|X_{i_{\max}}^\dagger - Y\|_2 < \varepsilon$ for some $Y \sim \pi$ independent of the initial point $X_0^\dagger$ with probability at least $1 - \delta$. Moreover UHMC requires at most $\tilde{O}(d^{1/4}\varepsilon^{-1/2}\sqrt{L_\infty}\log^{1/2}(\frac{1}{\delta}))$ gradient evaluations whenever $m, M, \omega = O(1)$, $r = \tilde{O}(d)$ and $L_\infty = \Omega(1)$.*

More generally, for arbitrary $m, M$ and $r$, we show (from a warm start with $\omega = O(1)$) that the number of gradient evaluations is $\tilde{O}(\max(d^{1/4}\kappa^{2.75}, r^{1/4}\sqrt{\tilde{L}_\infty}\kappa^{2.25})\tilde{\varepsilon}^{-1/2}\log^{\frac{1}{2}}(\frac{1}{\delta}))$, where $\tilde{L}_\infty := \frac{L_\infty}{\sqrt{M}}$ and $\kappa := \frac{M}{m}$ is the "condition number". From a cold start, under the additional Gaussian tail bound condition (Assumption 2), we show that the number of gradient evaluations is $\tilde{O}(\max(d^{1/4}\kappa^{3.5}, r^{1/4}\sqrt{\tilde{L}_\infty}(\kappa^{4.25} + \tilde{b}\kappa^{3.25}))\tilde{\varepsilon}^{-1/2})$, where $\tilde{\varepsilon} := \varepsilon/\sqrt{M}$ and $\tilde{b} := b/\sqrt{M}$. [1]

If $\kappa = O(1)$, $L_\infty = O(1)$ and $r = O(d)$, then our bound on the number of gradient evaluations is $O(d^{\frac{1}{4}}\tilde{\varepsilon}^{-\frac{1}{2}})$ from a warm start. To the best of our knowledge, our bounds are an improvement over all previous gradient evaluation bounds for sampling in this regime, which all have dimension dependence $\sqrt{d}$ or greater. Also note that while [33] obtains $O^*(d^{\frac{1}{4}})$ bounds in the special case of product distributions, unlike in [33] the condition number dependence in our bounds is polynomial.

We are especially interested in the regime where $d$ is large since the number of predictor variables in a statistical model is oftentimes very large [26, 22], and in many cases $\kappa$ and $L_\infty$ do not grow, or only grow relatively slowly, with the dimension. We state some concrete examples from Bayesian logistic regression of regimes where our gradient evaluation bounds are an improvement on the previous best bounds in the discussion after Theorem 2.

**Applications to logistic regression.** In Bayesian logistic "ridge" regression, one would like to sample from the target log-density

$$U(\theta) = \frac{1}{2}\theta^\top\Sigma^{-1}\theta - \sum_{i=1}^r \mathsf{Y}_i\log(F(\theta^\top\mathsf{X}_i)) + (1-\mathsf{Y}_i)\log(F(-\theta^\top\mathsf{X}_i)), \qquad (3)$$

where the data vectors $\mathsf{X}_1,\ldots\mathsf{X}_r \in \mathbb{R}^d$ are thought of as independent variables, the binary data $\mathsf{Y}_1,\ldots,\mathsf{Y}_r \in \{0,1\}$ are dependent variables, $F(s) := (e^{-s}+1)^{-1}$ is the logistic function, and $\Sigma$ is positive definite. We define the incoherence of the data as

$$\mathsf{inc}(\mathsf{X}_1,\ldots\mathsf{X}_r) := \max_{i\in[r]} \textstyle\sum_{j=1}^r |\mathsf{X}_i^\top\mathsf{X}_j|.$$

We bound the value of the infinity-Lipschitz constant in terms of the incoherence:

**Theorem 2** (**Regularity bounds for logistic regression**). *Let $U$ be the logistic regression target for $r > 0$ data vectors $\mathsf{X}_1,\ldots,\mathsf{X}_r$, and let $\mathsf{inc}(\mathsf{X}_1,\ldots,\mathsf{X}_r) \leq C$ for some $C > 0$. Then the infinity-norm Lipschitz assumption is satisfied with $L_\infty = \sqrt{C}$ and "bad" directions $\mathsf{u} = \left\{\frac{\mathsf{X}_i}{\|\mathsf{X}_i\|_2}\right\}_{i=1}^r$.*

The proof of Theorem 2 is given in the arXiv version [34]. In particular, when the incoherence is $\tilde{O}(1)$, the constant $L_\infty$ does not grow with dimension: This includes the separable case when the $\mathsf{X}_i$ vectors are orthogonal and have unit magnitude. It also includes, for instance, the non-separable case where $r = d$ and the $\mathsf{X}_i$ are unit vectors with the first $\sqrt{d}$ of the $\mathsf{X}_i$ vectors isotropically distributed, and the angle between any two of the remaining vectors is greater than $\frac{\pi}{2} - \frac{1}{d}$. In both these examples the number of gradient evaluations required by UHMC under a standard normal prior is $\tilde{O}(d^{\frac{1}{4}}\varepsilon^{-\frac{1}{2}})$, since $C, M, m^{-1}$, (and therefore $\kappa$ and $\tilde{L}_\infty$) are all $\tilde{O}(1)$. When all $r = d$ vectors are isotropically distributed, we have $\sqrt{\tilde{L}_\infty} = \tilde{O}(d^{\frac{1}{8}})$ and require $\tilde{O}(d^{\frac{3}{8}}\varepsilon^{-\frac{1}{2}})$ gradient evaluations. In all these examples we therefore obtain an improvement over the existing $\tilde{O}(\sqrt{d}\varepsilon^{-1})$ bounds of [9, 33].

We also provide bounds for $m$ and $M$, showing that our Lipschitz gradient and strong convexity assumptions are satisfied for $M = \lambda_{\max}\left(\Sigma^{-1} + \sum_{k=1}^r \mathsf{X}_k\mathsf{X}_k^\top\right)$ and $m = \lambda_{\min}(\Sigma^{-1})$, respectively. To obtain bounds in the cold start setting, we show Assumption 2 is satisfied with $b = 2\mathsf{inc}(\mathsf{X}_1/\sqrt{\|\mathsf{X}_1\|_2},\ldots\mathsf{X}_r/\sqrt{\|\mathsf{X}_r\|_2})$ if $\Sigma$ is a multiple of the identity (see arXiv version [34] for proofs).

## 5  Proof overview of Theorem 1

For simplicity of exposition, in this proof overview we consider the special case where the HMC algorithm is given a warm start and where $m, M = \Theta(1)$; the general case is proved in the arXiv version [34]. Recall that Algorithm 1 generates a Markov chain $X^\dagger$ which approximates the steps taken by the idealized HMC chain $X$. Since the idealized HMC chain $X$ was shown to mix quickly in [33], it is enough for us to bound the approximation error $\|X_i^\dagger - X_i\|_2 < \varepsilon$ for all $i \leq \mathcal{I}$, where, roughly, $\mathcal{I} = \Theta(\log\frac{1}{\varepsilon})$ is a bound on the mixing time of $X$, if each HMC trajectory is run for time $T = \Theta(1)$.

To prove the conjectured $O^*(d^{\frac{1}{4}})$ gradient evaluation bounds, it is enough to show that an error bound $\|X_i^\dagger - X_i\|_2 < \varepsilon$ holds for a numerical timestep-size $\eta = \Omega^*(d^{-\frac{1}{4}})$, since the HMC algorithm computes $\mathcal{I} = O^*(1)$ trajectories and for this choice of $\eta$ each trajectory takes $\frac{T}{\eta} = O^*(d^{\frac{1}{4}})$ gradient evaluations to compute. Our goal is therefore to show that the error $\|X_i^\dagger - X_i\|_2$ is bounded by $\varepsilon$ for all $i \leq \mathcal{I}$ whenever $\eta = O^*(d^{-\frac{1}{4}})$.

The structure of our proof is as follows: We begin by bounding the local error of the leapfrog integrator accumulated at each numerical step (Step 1). Then, we use the fact that (from a warm start) the momentum of the HMC trajectories is roughly $N(0, I_d)$ to show that the continuous HMC trajectories of the idealized chain $X$ are unlikely to travel quickly in any of the "bad" directions $u_i$ specified in Assumption 1 (Step 2). Specifically, we show that at every step $i$ with high probability the momentum of the HMC trajectories satisfy $\|p_t\|_{\infty,\mathsf{u}} = O(\log(d))$ at every time $t \in [0,T]$. We then combine steps 1 and 2 to show that the numerical HMC chain also does not travel too quickly in any of the "bad" directions, and use this fact together with our bounds in Step 2 to bound the global error of the numerical HMC trajectories (Step 3). Finally, we compute the value of $\eta$ needed to bound the error $\varepsilon$, and use this to bound the number of gradient evaluations.

Note that when proving bounds from a cold start we use the additional Assumption 2 instead of the invariant Gibbs distribution to control the behavior of the trajectories. Due to limited space, formal proofs are deferred to the arXiv version [34] of our paper.

**Step 1: Error bounds for leapfrog integrator.** In this subsection we show how to use Assumption 1 to bound the error of the leapfrog integrator. We are unaware of non-asymptotic second-order bounds for the leapfrog integrator, since the previous error bounds for leapfrog we are aware of only hold in the limit as the numerical step size $\eta$ goes to zero [3, 33, 1, 29, 24]. For this reason, we prove new non-asymptotic polynomial time bounds for leapfrog here. Key to our analysis is the observation that the position estimate $q_{j+1} = q_j + \eta p_j - \frac{1}{2}\eta^2 \nabla U(q_j)$ returned by the leapfrog integrator is exactly the second-order Taylor expansion for $q_\eta(q_j, p_j)$, and the momentum estimate $p_{j+1} := p_j - \frac{1}{2}\eta \nabla U(q_j) - \frac{1}{2}\eta \nabla U(q_{j+1})$ approximates (with third-order error) the second-order Taylor expansion for $p_\eta(q_j, p_j)$ in the following way:

$$p_{j+1} = p_j - \eta \nabla U(q_j) - \frac{1}{2}\eta^2 \frac{\nabla U(q_{j+1}) - \nabla U(q_j)}{\eta} \approx p_j - \eta \nabla U(q_j) - \frac{1}{2}\eta^2 H_{q_j} p_j. \qquad (4)$$

The error in the Taylor expansion is due to the fact that the Hessian $H_{q_j}$ is not constant over the trajectory. Roughly, we can use Assumption 1 to bound the error in the Hessian at each time $0 \le t \le \eta$:

$$\|(H_{q_t} - H_{q_0})p_t\|_2 \le L_\infty \|p_t\|_{\infty,u} \times \|(q_t - q_0)\|_{\infty,u} \sqrt{r} \approx tL_\infty \|p_t\|_{\infty,u}^2 \sqrt{r}. \qquad (5)$$

Using Equations (4) and (5), we get an error bound of roughly

$$\|p_{j+1} - p_t(q_j, p_j)\|_2 \le \eta^3 L_\infty \sup_{t \in [0,\eta]} \|p_t(q_j, p_j)\|_{\infty,u}^2 \sqrt{r}. \qquad (6)$$

Finally, we note that bounding the error for the position variable $\|q_{j+1} - q_t(q_j, p_j)\|_2$ can be accomplished using standard techniques which do not require Assumption 1.

**Step 2: Bounding $\|p_t\|_{\infty,u}$ for the idealized HMC chain.** Since the error bound of the leapfrog integrator depends crucially on $\|p_t\|_{\infty,u}$, our next task is to show that

$$\|p_t\|_{\infty,u} \le O(\text{polylog}(d, 1/\delta) + \omega) \qquad (7)$$

with high probability for the idealized HMC chain. To do so, we use the fact that (from a warm start) the distribution of the momentum at any point on an HMC trajectory is roughly $N(0, I_d)$. To show this, we would like to use the fact that the position and momentum of HMC trajectories from an idealized HMC chain started at the *stationary* distribution $\pi$, are jointly distributed according to the Gibbs distribution $\propto e^{-U(q_t)}e^{-\frac{1}{2}\|p_t\|_2^2}$ at any given time $t$.

**2a: Bounding $\|p_t\|_{\infty,u}$ from a stationary start** We first consider a copy $\tilde{Y}_i$ of the idealized chain started at the stationary distribution $\tilde{Y}_0 \sim \pi$, and show that the momentum $p_t$ of its trajectories satisfies $\|p_t\|_{\infty,u} = O(\log(d))$ at every time $t$ w.h.p. Since the $\tilde{Y}$ chain is started at the stationary distribution, it remains stationary distributed at every step and the position $q_t$ and momentum $p_t$ of its trajectories have Gibbs distribution $\propto e^{-U(q_t)}e^{-\frac{1}{2}\|p_t\|_2^2}$ at any fixed time $t$. Using the fact that for every bad direction $u_i$, $|u_i^\top p_t|$ is chi-distributed with 1 degree of freedom, we apply the Hanson-Wright inequality together with a union bound to show that $\|u_i^\top p_t\|_{\infty,u} = O(\log(d/\xi))$ at any fixed time $t$ with probability at least $1 - \xi$ for any $\xi > 0$. However, our goal is to bound $\|u_i^\top p_t\|_{\infty,u}$ simultaneously at every time $t$, not just at a fixed time. Unfortunately, the trajectories are continuous paths, so we cannot directly apply a union bound to obtain a bound at every $t$. To get around this problem, we consider $\mathcal{J} = \text{poly}(\kappa, d)$ equally spaced timepoints on the interval $[0, T]$, and apply a union bound to show that $\|u_i^\top p_t\|_{\infty,u} = \text{polylog}(d, 1/\delta)$ with probability at least $1 - \delta$. We then use the "conservation of energy" property to bound the Euclidean norm of the momentum at every time on the trajectory, implying that the position and momentum do not change by more that $O(1)$ inside each time interval of length $\frac{1}{\mathcal{J}}$. This in turn implies that $\|p_t\|_{\infty,u} = \text{polylog}(d, 1/\delta)$ at every time $t \in [0, T]$.

**2b: Bounding $\|p_t\|_{\infty,u}$ from a warm start** Unfortunately, we cannot apply our results of step 2a directly since we are only assuming that $X_0$ has a warm start, not a stationary start. That is, we only assume that $\|X_0 - \tilde{Y}_0\|_2 < \omega$ for some $\omega > 0$, where $\tilde{Y}_0 \sim \pi$ is at the stationary distribution. To show that the trajectories of our warm-started chain also approximately satisfy this Gibbs distribution property, we couple the two copies $X$ and $\tilde{Y}$ of the idealized HMC chain by defining the $\tilde{Y}$ chain using the update rule $\tilde{Y}_{i+1} = q_T(\tilde{Y}_i, \mathbf{p}_i)$ with the same sequence of initial momenta $\mathbf{p}_1, \mathbf{p}_2, \ldots$ that were used to define the $X$ chain. Using the fact that the trajectories share the same initial momentum $\mathbf{p}_i$ at every step, we show that at every continuous time $t \in [0, T]$ the Euclidean distance between the position and momentum of the trajectories of the two chains remains bounded by $\omega$. We therefore have $\|p_t(X_i, \mathbf{p}_i) - p_t(\tilde{Y}_i, \mathbf{p}_i)\|_{\infty,u} \le \|p_t(X_i, \mathbf{p}_i) - p_t(\tilde{Y}_i, \mathbf{p}_i)\|_2 \le \omega$.

**Step 3: Bounding the global error and the number of gradient evaluations.** So far, we have shown that the trajectories of the idealized HMC chain $X$ satisfy a bound on $\|p_t\|_{\infty,\mathsf{u}}$ (Equation (7)). If we can extend this bound to the numerical chain, we can apply it to Inequality (6) to show that the error at each step is $O(\eta^3 L_\infty \sqrt{r})$ w.h.p. To bound the global error, we use roughly the following inductive argument: inductively assume that the errors $\|\mathsf{q}_j - q_{j\eta}(X_i, \mathbf{p}_i)\|_2$ and $\|\mathsf{p}_j - p_{j\eta}(X_i, \mathbf{p}_i)\|_2$ at numerical step $j$ are bounded by roughly $j\eta \times \varepsilon$. This implies that

$$\|\mathsf{p}_j\|_{\infty,\mathsf{u}} \le \|p_{j\eta}(X_i, \mathbf{p}_i)\|_{\infty,\mathsf{u}} + j\eta\varepsilon \overset{\text{Eq. 7}}{=} \text{polylog}(d, 1/\delta) + \omega. \tag{8}$$

Then one can use similar "conservation of energy" arguments as in the previous section to show that $\|p_t(\mathsf{q}_j, \mathsf{p}_j)\|_{\infty,\mathsf{u}} = \text{polylog}(d, 1/\delta)$ over the short time interval $t \in [0, \eta]$. Plugging this bound into Inequality (6) allows us to bound the error accumulated at step $j$ by $O(\eta^3 L_\infty \sqrt{r})$, implying that the inductive assumption also holds for step $j + 1$.

After $\frac{T}{\eta}$ numerical steps, the global error of each trajectory is therefore bounded by $T \times \eta^2 L_\infty \sqrt{r}$. The error at step $i$ is bounded by $i \times T\eta^2 L_\infty \sqrt{r}$. Finally, we conclude that $\|X_i^\dagger - X_i\|_2 < \varepsilon$ for $i \le \mathcal{I} = O(\log(\frac{1}{\varepsilon}))$ whenever $\eta^{-1} = \tilde{\Theta}(r^{\frac{1}{4}}\sqrt{\tilde{L}_\infty}\varepsilon^{-\frac{1}{2}}T)$. Since the algorithm uses a total of $\frac{T}{\eta}$ numerical steps to compute $X_i$ for $i \le \mathcal{I}$, for $T = \Theta(1)$ the number of gradient evaluations is roughly $\tilde{O}(r^{\frac{1}{4}}\sqrt{\tilde{L}_\infty}\varepsilon^{-\frac{1}{2}})$. When $r = O(d)$, the number of gradient evaluations is roughly $\tilde{O}(d^{\frac{1}{4}}\sqrt{\tilde{L}_\infty}\varepsilon^{-\frac{1}{2}})$.

**Remark 2.** *More generally, we can consider Hamiltonian trajectories with Hamiltonian $\mathcal{H}(q, p) = U(q) + \frac{1}{2}p^\top \Omega p$, where $\Omega$ is called the mass matrix. In practice, one tunes the algorithm parameters by using a mass matrix $\Omega = cI_d$ for some constant $c > 0$ and by choosing an appropriate integration time $T$ (as well as choosing other parameters such as numerical step-size). Since using a mass matrix of the form $\Omega = cI_d$ for some constant $c$ is equivalent to rescaling $U$ by a constant factor and tuning the integration time $T$, we analyze the case $\Omega = I_d$ and then "tune" our algorithm by setting $T = \frac{\sqrt{m}}{6}$, $M = 1$ and $\kappa = \frac{1}{m}$ to determine the number of gradient evaluations. (A more general mass matrix $\Omega$ is equivalent to applying a pre-conditioner on $U$.)*

# 6 Simulations

## 6.1 Accuracy and autocorrelation time of Unadjusted HMC

The purpose of our first set of simulations is to show that in practical situations analyzed in this paper the unadjusted HMC algorithm (UHMC) is competitive with other popular sampling algorithms in terms of both accuracy and in terms of the number of gradient evaluations required. We compare UHMC to Metropolis-adjusted HMC (MHMC) [14], the Metropolis-adjusted Langevin algorithm (MALA) [41] and the unadjusted Langevin algorithm (ULA) [41]. All simulations were implemented on MATLAB (see the GitHub repository `https://github.com/mangoubi/HMC` for our MATLAB code used to implement these algorithms).

We consider the setting of Bayesian logistic regression with standard normal prior, with synthetic "independent variable" data vectors generated as $\mathbf{X}_i = \frac{Z_i}{\|Z_i\|_2}$ for $Z_1, \dots, Z_r \sim N(0, I_d)$ iid, for dimension $d = 1000$ and $r = d$. To generate the synthetic "dependent variable" binary data, a vector $\beta = (\beta_1, \dots, \beta_d)$ of regression coefficients was first generated as $\beta = \frac{W}{\|W\|_2}$ where $W \sim N(0, I_d)$. The binary dependent variable synthetic data $\mathsf{Y}_1, \dots, \mathsf{Y}_d$ were then generated as independent Bernoulli random variables, setting $\mathsf{Y}_i = 1$ with probability $\frac{1}{1+e^{-\beta^\top \mathbf{X}_i}}$ and $\mathsf{Y}_i = 0$ otherwise. Each Markov chain was initialized at a point $X_0$ chosen randomly as $X_0 \sim N(0, I_d)$.

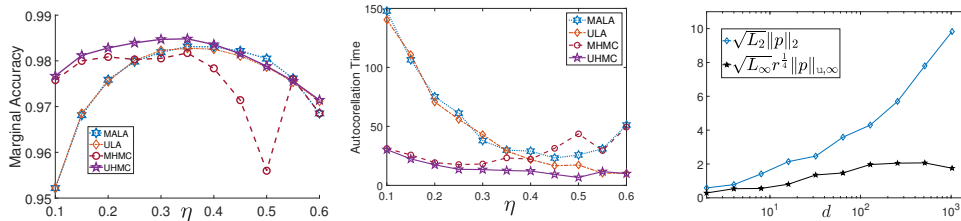

Figure 1: Marginal accuracy (left) and autocorrelation time (middle) vs. numerical step size $\eta$ for MALA, ULA, MHMC, and UHMC. The log plot (right) compares an estimate for the quantity used to obtain second-order numerical error bounds using the Euclidean Lipschitz and infinity-norm Lipschitz constants, at different $d$.

To compare the accuracy, we computed the "marginal accuracy" (MA) of the samples generated by each chain over a fixed number (50,000) of numerical steps for different step sizes $\eta$ in the

interval $[0.1, 0.6]$ (Figure 1, left). Among all four of the algorithms, we found that UHMC had the highest accuracy at the accuracy-optimizing step size (the accuracy-optimizing step size was $\eta = 0.35$ for UHMC). To compare the runtime we computed the autocorrelation time of the samples for a test function $f(x) = \|x\|_1$.[2] We found that the autocorrelation time of UHMC was fastest at the autocorrelation time-optimizing step size (the autocorrelation time-optimizing step size was $\eta = 0.5$ for UHMC) (Figure 1, middle). When running UHMC and MHMC, we used a trajectory time $T$ equal to $\frac{\pi}{3}$, rounded down to the nearest multiple of $\eta$.

**Remark 3.** *The marginal accuracy is used as a heuristic to compare accuracy of samplers (see e.g. [18], [20] and [10]). The marginal accuracy between the measure $\mu$ of a sample and the target $\pi$ is $MA(\mu, \pi) := 1 - \frac{1}{2d} \sum_{i=1}^{d} \|\mu_i - \pi_i\|_{\text{TV}}$, where $\mu_i$ and $\pi_i$ are the marginal distributions of $\mu$ and $\pi$ for the coordinate $x_i$. Since MALA is known to sample from the correct stationary distribution and is geometrically ergodic for the class of distributions analyzed in this paper, we used the samples generated after running MALA for a very long time ($10^6$ steps) to obtain a more accurate approximation for $\pi$ as a benchmark with which to compare the sampling accuracy of the four different algorithms when run for a much shorter amount of time ($50,000$ numerical steps).*

### 6.2 Comparing Euclidean and infinity-norm Lipschitz conditions

The goal of our second set of simulations was to compare the optimal values of the usual Euclidean Lipschitz Hessian constant $L_2$ to the constant $L_\infty$ from our infinity-norm Lipschitz condition of Assumption 1. We performed this comparison for the logistic regression example of the previous simulation with synthetic data generated in the same way, but for different values of $d$, with $r = d$. The optimal values of $L_2$ and $L_\infty$ are $L_2 = \sup_{x,y,v \in \mathbb{R}^d} \frac{1}{\|y-x\|_2 \|v\|_2} \|(H_y - H_x)v\|_2$ and $L_\infty = \sup_{x,y,v \in \mathbb{R}^d} \frac{1}{\sqrt{r}\|y-x\|_{\infty,\mathsf{u}} \|v\|_{\infty,\mathsf{u}}} \|(H_y - H_x)v\|_2$, with "sup" taken over points where function is defined.

At each value of $d$ we used MATLAB's "fminunc" function to search for the optimal values of $L_2$ and $L_\infty$. Recall that to bound the error of a numerical integrator with momentum $p_t$, one may use one of the two quantities $\sqrt{L_2}\|p_t\|_2$ and $\sqrt{L_\infty} r^{\frac{1}{4}} \|p_t\|_{\infty,\mathsf{u}}$. We plot the median value of these quantities for random momenta $p_t \sim N(0, I_d)$ (Figure 1, right). Our results show that the median of $\sqrt{L_2}\|p_t\|_2$ increases with $d$ at a faster rate than the median of $\sqrt{L_\infty} r^{\frac{1}{4}} \|p_t\|_{\infty,\mathsf{u}}$ over the interval $d \in [1, 1000]$. This suggests that bounds based on our infinity-norm Lipschitz condition can be much tighter for distributions used in practice than bounds based on the usual Euclidean Lipschitz condition.

## 7 Conclusions and future directions

In this paper, we show that the conjecture of [11], which says that HMC requires $O^*(d^{1/4})$ gradient evaluations, is true when sampling from strongly log-concave targets satisfying weak regularity properties associated with the input data. In doing so, we introduce a new regularity property for the Hessian (Assumption 1) that is much weaker than the Lipschitz Hessian property, and show that for a class of functions arising in statistics and machine learning this property holds for natural conditions on the data. One future direction is to further weaken Assumption 1, which says that the Hessian does not change too quickly in all but a few fixed bad directions, by instead allowing these directions to vary based on the position $x$. Our simulations show that UHMC is competitive with MHMC on synthetic data that satisfies our regularity assumption. Further, we show that the constant in our regularity assumption grows much more slowly with the dimension than the Euclidean Lipschitz constant of the Hessian. It would also be interesting to extend our results to non-logconcave targets, and to $k$th-order numerical implementations of generalizations of HMC, such as RHMC.

**Bounds for MHMC**  Another open problem is to show tight gradient evaluation bounds for MHMC. Since the Metropolis-adjusted HMC Markov chain preserves the stationary distribution exactly, it should be possible to show that the number of gradient evaluations is polylogarithmic in $\varepsilon^{-1}$, improving on the number of gradient evaluations required by unadjusted HMC which grows like $\varepsilon^{-\frac{1}{2}}$. Unfortunately, the probablistic coupling approach used in our current paper is unlikely to work for MHMC, since Metropolis "accept/reject" steps tend to break the coupling of the two Markov chains if the chains have different acceptance probabilities, causing one chain to accept its proposal while the other rejects the proposal. An alternative approach might be to use a proof based on the conductance method (see e.g. [43]). Unlike the coupling approach, the conductance method is compatible with Metropolis "accept/reject" steps.

## Footnotes

[1]To obtain bounds from a warm start, we run UHMC with parameters $T = (6\sqrt{M\kappa})^{-1}$, $i_{\max} = \Theta(\frac{1}{mT^2})$, and $\eta = \Theta(\min\{d^{-1/4}\kappa^{-1.25}, r^{-1/4}\tilde{L}_\infty^{-1/2}\kappa^{-0.75}\}\tilde{\varepsilon}^{\frac{1}{2}}M^{-\frac{1}{2}}\log^{-\frac{1}{2}}(\frac{1}{\delta}))$. From a cold start, we run UHMC with parameters $T = (6\sqrt{M\kappa})^{-1}$, $i_{\max} = \Theta(\frac{1}{mT^2})$, and $\eta = \Theta(\min\{d^{-\frac{1}{4}}\kappa^{-2}, r^{-1/4}\tilde{L}_\infty^{-1/2}(\kappa^{2.75} + \tilde{b}\kappa^{1.75})^{-1}\}\tilde{\varepsilon}^{1/2}M^{-1/2})$.

[2]The autocorrelation time can be estimated as $1 + 2\sum_{s=1}^{s_{\max}} \rho_s$, where $\rho_s$ is the autocorrelation at lag $s$ for some large $s_{\max}$[23, 6].

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
