[Reviews · NeurIPS 2018]

Reviewer 1



The authors establish that the second-order leapfrog unadjusted HMC algorithm runs in d^{1/4} steps. This result had been conjectured but had never been established. It is a long and theoretical paper, it is strictly impossible to verify all the technical details of a difficult proof of more than 20 pages. I have only superficially checked the supplementary materials. The few fragments of proofs I read were convincing and correct. This is a very theoretical result but it explains the superiority of HMC methods and the feasibility of Bayesian methods (at least for convex potentials) for high-dimensional problem.

Reviewer 2



Added after author response: I'm now happier with the simulations after reading the author response. I'm still concerned that the actual proof is 30 pages, and hence not in the paper, and not checkable in the time frame of a NIPS review. ---- This paper (2949) proves results (in supplementary information, with summaries in the paper) on how the computation time of Hamiltonian Monte Carlo sampling grows with increasing dimensionality, d. There have long been heuristic arguments that the number of gradient evaluations required to sample a nearly independent point should usually scale as d^1/4. (So assuming time growing as d per gradient evaluation, total run time scales as d^5/4.) This paper contributes to putting this on a firmer foundation, for a wider class of distributions, including some of practical interest. The main focus of the paper is the unadjusted HMC algorithm, in which the accept/reject test is omitted. When the dynamics is simulated with the leapfrog method, the distribution sampled from when there is no accept/reject test is not the one desired, but becomes closer to the desired distribution as the leapfrog stepsize is reduced. The point of the results in this paper is to investigate how fast, in order to maintain some desired accuracy bound. the stepsize needs to be made smaller (with correspondingly longer computation time) as d is made bigger. Some more discussion of how it might (or might not) be possible to connect this result to a similar result for standard HMC with an accept/reject test would be interesting. This is presently only briefly hinted at at the end of the paper. Some simulations are presented of HMC and other methods for Bayesian logistic regression. These could be improved, as discussed below. Overall, this is a useful contribution, which however does not fit entirely comfortably within NIPS. The supplementary information with the actual proof is 30 pages long. A final NIPS paper would presumably be able to reference this as archived somewhere else. It is of interest to have a summary paper to expose progress in this area to a wider audience. Remarks on Section 6 (Simulations): Some aspects of the simulations are not adequately described. What was the value of r? What values for the regression coefficients were used in generating the data? (Or were many values drawn from the prior used, with the results shown being averages?) What were start states for the chains? With a standard normal prior on 1000 coefficients, and standard normal inputs, the input to the logistic function is very likely to be far from zero, so the probability of the response being 1 will be very close to 0 or 1. This is not typical of most real logistic regression problems. It is odd to say things like (lines #335-336), "the autocorrelation time of UHMC was fastest over the interval [0.15,0.5], while ULA was fastest at eta=0.55", as if a value of eta was somehow externally imposed, rather than being freely choosable. Detailed comments: #25: I think i times eta^2 should be i^2 times eta. #44, #72-74: HMC (with leapfrog integration) applied to Gaussian distributions was also analysed pretty completely in the following paper: Kennedy, A. D. and Pendleton, B. (1991) "Acceptances and autocorrelations in hybrid Mone Carlo", Nuclear Physics B (Proc. Suppl.), vol 20, pp. 118-121. Note that a leapfrog step with a Gaussian target distribution is a linear map, making the analysis not too difficult. #99: "[34,2]" should probably be "[35,2]". Reference [41] is a garbled combination of two unrelated references.

Reviewer 3



This paper looks at the "running time" of HMC. It seems one of the prime contributions of this work is a relaxation of a condition given in prior work (which this reviewer has not read) to accommodate a more general regularity structure on the target. Namely, Assumption 1 in the present work is a kind of Lipschitz assumption on the Hessian but based on some apparently weaker norm. Under this condition, and importantly strongly concave targets, the authors look at the number of "gradient evaluations" required in HMC. I think this is interesting from the theoretical point of view, and perhaps useful - albeit the nature of the problems addressed are not so interesting alone and thus the practical impact of this work is limited I would guess. I have some concerns and I don't think the paper is really ready for publication in NIPS. These are minor in some sense. Firstly, the idea of "run time" from the start is pretty vague. It seems it is only in Theorem 1 that running time is defined explicitly as "the number gradient evaluations". Even this is perhaps not completely transparent. The Algorithm 1 has a fixed running time determined by i_{max}, T, \eta, and it is independent of d in terms of the number of gradient evaluations. So it seems the notation between this and, for example, Theorem 1, inconsistent. Line 107 - why do I have only p_1 and p_2, I am confused? What is the "^\prime" symbol, surely it is used for multiple purposes or something is wrong here? What is the norm used in the Theorem 1 to measure X_{i_{max}} and Y? Perhaps this is mentioned somewhere but I missed it. Same question in the definition of a warm start and in the Section 5. This seems rather critical? And then what is a "high probability" in Theorem 1. I don't really follow this theorem in a rigorous sense nor its proof outline - I think there are some problems with a lack of defining an appropriate distance here. I didn't read the supplementary material. Personally, for this reviewer, the formatting with a lot of "in-line" notation made the paper a little hard and frustrating to read. But this is certainly not a criticism for rejection - rather just a comment that may aid the authors in presenting a revision. In summary, I think there is a lot of potential with this result and it would be of theoretical interest to people working in this particular direction in this area. However, from my reading I didn't get the feeling it was polished enough and perhaps suitable enough for NIPS although I am happy to be corrected. ---- Added after author rebuttal: I am happy with the changes. I don't think you need the "_2" subscript on the norm if you define it - previously you had the Euclidean norm with "_2" in some places and someplace without. For notation, there are a lot of places I don't see the notation as necessary or even understandable. For example, the \dagger in Algorithm 1 seems completely unnecessary. As does the \tilde in the warm start definition. I don't follow the \tilde{O} notation - I guess this is to distinguish it from O^\star which only captures the dimension terms, but then just plain O( ) notation is also used so the \tilde{O} is unknown to me (perhaps its separate but standard notation I should know?). Anyway, these are just minor issues really; but I saw a lot of probably unnecessary notation which could probably be worked on in later versions (e.g. the superscript notation later on in the leapfrog integrator section (7 in the arXiv) is a bit much. My main concern is the length and the suitability for NIPS in terms of acceptance. I like the idea of the result. But unadjusted HMC is not really a practically chosen algorithm. And the length of the supplementary material - which I tried in the rebuttal period to follow but had to stop due to time constraints - is too long. I don't believe supplementary material's purpose is to contain the bulk of an article, which is the case when the main contribution of the article is a theorem and the supplementary contains the proof. So while I think the paper has merit, I think it needs to benefit from a proper review of the entirety of the technical details in a more suitable venue (e.g. some probability journal). I note, for example, similar studies on unadjusted Langevin appear in AAP for example.